# Ionic Liquid-Based Ultrasonic-Assisted Extraction Coupled with HPLC and Artificial Neural Network Analysis for *Ganoderma lucidum*

**DOI:** 10.3390/molecules25061309

**Published:** 2020-03-13

**Authors:** Changqin Li, Yiping Cui, Jie Lu, Cunyu Liu, Sitan Chen, Changyang Ma, Zhenhua Liu, Jinmei Wang, Wenyi Kang

**Affiliations:** 1National R & D Center for Edible Fungus Processing Technology, Henan University, Kaifeng 475004, China; lcq@vip.henu.edu.cn (C.L.); cuiyiping@vip.henu.edu.cn (Y.C.); Lujie980091180@163.com (J.L.); liucunyu@vip.henu.edu.cn (C.L.); chensitan1995@163.com (S.C.); macaya1024@vip.henu.edu.cn (C.M.); 2Kaifeng Key Laboratory of Functional Components in Health Food, Kaifeng 475004, China; 3Joint International Research Laboratory of Food & Medicine Resource Function, Henan Province, Kaifeng 475004, China

**Keywords:** *Ganoderma lucidum*, ionic liquid (ILs), ultrasonic-assisted method, HPLC, artificial neural network

## Abstract

*Ganoderma lucidum* is widely used in traditional Chinese medicine (TCM). Ganoderic acid A and D are the main bioactive components with anticancer effects in *G. lucidum*. To obtain the maximum content of two compounds from *G. lucidum*, a novel extraction method, an ionic liquid-based ultrasonic-assisted method (ILUAE) was established. Ionic liquids (ILs) of different types and parameters, including the concentration of ILs, ultrasonic power, ultrasonic time, rotational speed, solid–liquid ratio, were optimized by the orthogonal experiment and variance analysis. Under these optimal conditions, the total extraction yield of the two compounds in *G. lucidum* was 3.31 mg/g, which is 36.21% higher than that of the traditional solvent extraction method. Subsequently, an artificial neural network (ANN) was developed to model the performance of the total extraction yield. The Levenberg–Marquardt back propagation algorithm with the sigmoid transfer function (logsig) at the hidden layer and a linear transfer function (purelin) at the output layer were used. Results showed that single hidden layer with 9 neurons presented the best values for the mean squared error (MSE) and the correlation coefficient (R), with respectively corresponding values of 0.09622 and 0.93332.

## 1. Introduction

As a kind of macro fungi, mushrooms have been cultivated and consumed by humans because of its attractive sensory characteristics rich nutritional components, multiple functional activities, and controllable cultivation conditions [1]. The medicinal mushrooms mainly include *Ganoderma lucidum*, *Cordyceps sinensis,* and *Poria Cocos*.

The dry fruiting body of *Ganoderma lucidum* (Curtis: Fr.) P. Karst has been used for over 1000 years [2,3] as Traditional Chinese Medicine (TCM), which is a widely used as a dietary supplement and medicinal herb in China and other eastern countries [4]. It is one of the most highly used medicinal fungi around the world [4]. Studies have shown that *G. lucidum* has the pharmacological effect of immunoregulation [5], antitumor [6,7], liver protection [8], and blood glucose reduction [9]. Ganoderic acid A and D (Figure 1) are the main triterpenoids in *G. lucidum* [4]. It is reported that Ganoderic acid A has the effects of analgesic [10], antioxidant [11], liver protection [12], and anticancer [13,14,15,16,17] effect. Furthermore, it has been found that Ganoderic acid A has significant antitumor effect on human osteoma cells [13], lymphoma cells [14], meningioma cells [15], and breast cancer cells [17] and can also improve the chemosensitivity of HepG2 cells to platinum [16]. Therefore, ganoderic acid A may be a candidate drug for new anticancer drugs. Ganoderic acid D can inhibit the proliferation of cervical cancer cells [18], regulate energy metabolism of colon cancer cells through irt3/cypd signal pathway, and then inhibit colon cancer [19].

The extraction of effective components has been widely reported [20,21,22,23]. However, the extraction with traditional organic solvent consumes a lot of energy and pollutes the environment.

Ionic liquid (IL) is a new solvent, which is composed of organic cation and inorganic or organic anion [24]. As another kind of green solvent after water and supercritical carbon dioxide, ionic liquid has unique physical and chemical properties, such as negligible vapor pressure, good thermal and chemical stability, high solubility to organic and inorganic substances, and modifiable chemical structure [25,26,27,28]. As a green environmental solvent, ionic liquid has been widely used to extract active ingredients from natural products [29].

Artificial neural network (ANNs) is an information processing system based on imitating brain function, which can quickly and accurately realize the simulation and prediction of data information [30]. ANNs reduces the time of experiment and model development without the mathematical description of the phenomena involved in the process [31,32]. Several works showed that ANNs has been widely used in the optimization of chemical and physical process parameters [33,34]. BP neural network, also known as back propagation (BP) neural network, is a multilayer feedforward neural network in the artificial neural network, which is the most representative and extensive.

To our best knowledge, the extraction of triterpenoids from *G. lucidum* by ionic liquid assisted extraction has not been reported. So, ionic liquid ultrasonic assisted as extract technology, and BP neural network as analysis method were used to simultaneously determine the content of ganoderma acid A and D in *G. lucidum* by HPLC to obtain the optimal extraction process parameters in this paper.

## 2. Results and Discussion

### 2.1. Linear Relationship

For preparing standard sample solutions, various amounts of ganoderic acid A and D were dissolved in methanol to yield their stock solutions, respectively. Corresponding calibration curves for ganoderic acid A and D were Y = 1184987.20X + 23402.57, (*r* = 0.9999) and Y = 1231665.48X − 28013.61, (*r* = 0.9997), respectively. Ganoderic acid A and D showed a good linearity in the ranges of 0.062~46.5 (μg/mL) and 0.0218~16.35 (μg/mL), respectively. The limit of detection (LODs, based on signal-to-noise ratio of 3, S/N = 3) and the limit of quantification (LOQs, based on signal-to-noise ratio of 10, S/N = 10) of ganoderic acid A were 10.1 and 3.8 ng, respectively; LOD and LOQ of ganoderic acid D were 9.6 and 2.4 ng, respectively.

### 2.2. Selection Period of Ionic Liquids (ILs)

Selecting the appropriate solvent is of great importance in order to obtain a satisfactory extraction efficiency of the target compounds. First, three different traditional solvents: water, 70% ethanol, and methanol were selected for extraction. These solvents can extract the target compounds, and methanol has the highest extraction yield of the target compounds (Figure 1). Because ionic liquids are mostly viscous liquids, it is particularly important to choose a suitable solvent to dissolve the ionic liquid. Then, it can further improve the extraction yield of target components in the sample.

The physical and chemical properties of ionic liquids are determined by their structure, which will significantly affect the extraction yield of target compounds [35]. Therefore, selecting the appropriate extractant is the key step for effective extraction. Five kinds of ionic liquids: [BMIM] Br, [BMIM] BF_4_, [BMIM] PF_6_, [HMIM] Br, [OMIM] Br with different anions and cations were selected in this study.

In Figure 2, when the cation structure of the ionic liquid was the same, the extraction effect of ionic liquids with different anions was different, and the extraction rate was Br^−^ > BF 4^−^ > PF6^−^. These anions may interact with the hydroxyl groups of the target compounds, especially the hydrogen bond, π–π conjugation, and ion/charge forces, which are conducive to the dissolution and extraction of the target compound [36]. On the other hand, the ion radius of Br^−^ is large, and it is easy to polarize and deform under the action of external electric field. The induced dipole moment enhances the ability of the solution system to absorb the energy of external magnetic field. Therefore, the extraction effect of the ionic liquid containing Br^−^ is better than that of the other two anions.

In Figure 3, [BMIM] Br, [HMIM] Brand [OMIM] Br had the same anions, but the alkyl chain length of the cation increased from hexyl to octyl. It showed that the length of the alkyl chain had a great influence on the extraction rate. The extraction rate was [HMIM]^+^> [BMIM]^+^> [OMIM]^+^. The extraction rate first increased and then decreased with the increase of alkyl chain length. This may be the longer the carbon chain, the stronger the hydrophobicity; the stronger the van der Waals force, and the higher the extraction rate. However, the increase of the carbon chain will increase the viscosity of the ionic liquid, hinder the mass transfer, and reduce the extraction rate.

Therefore, [HMIM] Br/methanol was selected as the best extraction solvent. After consulting relevant literature, Shi [37] selected [HMIM]PF6/70% EtOH to extract the compounds from *Psoralea Fructus*, because of the highest extraction efficiency. Tan [38] used MILs[C_4_mim]FeCl_3_Br along methanol for extracting paclitaxel from Taxus species. Different types of compounds need different types of ionic liquids to achieve the highest extraction yield, so it is necessary to select the best extraction solvent.

#### 2.2.1. Selection of Concentration of ILs

The concentration of ionic liquid is an important parameter affecting the extraction rate. Different concentrations have different extraction and dissolution capacities [39]. In Figure 4, when the concentration of [HMIM] Br increased from 0.8 to 1.2 M, the extraction yield increased, but when the concentration of [HMIM] Br was greater than that of 1.2 M, the extraction yield decreased. When the concentration of [HMIM] Br is at a low level, the solubility of the ionic liquid to the target compound increases with the increase of the concentration; when the concentration of [HMIM] Br is greater than a certain value, the viscosity of the ionic liquid increases, making it difficult for the solvent to enter the sample, and the target compound is not easy to diffuse. Therefore, the optimal concentration of 1.2 M was selected.

#### 2.2.2. Selection of Ultrasonic Power

Different ultrasonic power leads to different extraction rate. Wang [40] chose 300 W as the best ultrasonic power. A strong mechanical effect (such as shear) was produced by ultrasound, causing cell rupture [41]. Effective cell division and mass transfer are the main factors leading to the increase of extraction efficiency. In Figure 5, when the ultrasonic power was less than 350 W, the extraction rate of the target compound increased with the increase of ultrasonic power. When the power was greater than 350 W, the extraction yield decreased instead. It may be that the enhancement of ultrasonic power could quickly destroy plant cells and accelerate the dissolution of target compounds under low power. However, high ultrasonic power may destroy target compounds and decrease the extraction rate. Therefore, 350 W was selected as the best ultrasonic power.

#### 2.2.3. Selection of Ultrasonic Time

Ultrasound time has a certain impact on the extraction rate. Huang and Su [42,43] have screened the extraction time for subsequent experiments. Meanwhile, In Figure 6, the extraction yield of the target compounds increased with the prolongation of ultrasonic time before 20 min. The extraction yield reached the maximum when the ultrasonic time was 20 min, and the extraction yield of the target compounds presented a downward trend with the increase of time. This may be because it took a certain time for the penetration of ionic liquids to enter into plant tissues and dissolve the target compounds. When the extraction time was too long, the structures of ionic liquids and target compounds were destroyed. Thus, 20 min was selected for the follow-up experiment.

#### 2.2.4. Selection of Rotational Speed

The highest extraction rate was obtained by selecting the best centrifugal speed [44]. Proper rotational speed can not only improve work efficiency but also reduce energy consumption and save money. Under the above optimized conditions, the influence of rotational speed on the extraction yield was investigated. In Figure 7, when the rotational speed was 4000 r/min, the extraction yield of the target compounds reached the maximum. After comprehensive consideration, 4000 r/min was selected for subsequent operation.

#### 2.2.5. Selection of Solid–Liquid Ratio

The solid–liquid ratio is an important factor affecting the extraction yield. If the solid–liquid ratio is too small, the sample does not contact the extractant completely, and the extraction is insufficient. If the solid–liquid ratio is too large and the solvent is too much, the solvent will be wasted, and the cost will be increased. Many studies have considered the factor of solid–liquid ratio [37,43]. In Figure 8, when the solid–liquid ratio was 1:20 g/mL, the extraction yield of the target compounds reached the maximum. This may be because the increase of solvent volume could fully contact with the sample and dissolve the target compound. With the increase of solvent volume, the viscosity of extraction solvent and the yield of non-target-compounds may be increased, resulting in incomplete extraction of the target compounds.

### 2.3. Optimization Extraction Process in *Ganoderma lucidum*

Various parameters play an important role in the optimization of the experimental conditions to establish the method of solvent extraction. The levels of each factor were selected according to the above results of the single factor (in Table 1). Through the SPSS 19.0, the blank column design orthogonal test was added, and the optimum extraction conditions were tested with the extraction yield as the index.

The analysis results of orthogonal test, performed by statistical software SPSS 19.0, are presented in Table 2 and Table 3.

#### 2.3.1. The Results of the Intuitionistic Analysis

In Table 2, 5 factors (ILs concentration, ultrasonic power, ultrasonic time, rotational speed, and solid–liquid ratio) had great influence on the experimental results. Among them, rotational speed was the most important parameter.

The factors influencing the extraction yield were listed in a decreasing order as follows: D > B > C > E > A according to their R values.

However, the estimate of error cannot be calculated by intuitionistic analysis, which cannot accurately reflect the experimental error or a substantial change between the levels [45]. Therefore, in order to fully and more accurately express the experimental results, further analysis is needed.

#### 2.3.2. The Results of the Variance Analysis

With the extraction yield as the index, the variance analysis was carried out by SPSS 19.0 software. In Table 3, the results showed that the D factor (rotational speed) was significant. The result was consistent with the visual analysis. A_3_B_3_C_2_D_2_E_3_ was identified as the extraction process as follows: the optimal IL concentration, 1.4 M; ultrasonic extraction power, 400 W; ultrasonic extraction time, 20 min; rotational speed, 4000 r/min; solid–liquid ratio, 1:33.3 g/mL. This optimal process was repeated three times, and the extraction yield of the target compound was 3.31 mg/g (*n* = 3).

#### 2.3.3. Comparison between IL-UAE Approach and the Traditional Methods

Under the optimal conditions by [HMIM] Br/methanol, the average contents of the target compounds was 3.31 (mg/g) (*n* = 3), while the average contents of the target compounds obtained by traditional solvent–methanol extracting was 2.43 (mg/g) (*n* = 3). The results showed that the method of IL-UAE increased the extract rate 36.21% compared with the traditional method, and the extraction process was optimized by orthogonal test.

### 2.4. ANN Model Development

BP neural network is the most typical in nonlinear function fitting. As long as the appropriate number of neurons is selected in the hidden layer, a nonlinear function can be approximated with any accuracy. Therefore, three-layer neural network was used in this experiment. MATLAB software was used to complete the analysis and modeling of test data.

The ANN model development was based the data of single factor experiments and orthogonal experiment. The input variables were the concentration of selected IL, the ultrasonic power, the ultrasonic time, the rotational speed, and solid–liquid ratio. The extraction yield was used as an output variable.

The neurons number in the hidden layer affects the performance of the ANN model. Indeed, a very small number of neurons causes underfitting, and on the contrary, an excess of neurons leads to overfitting by raising the complexity of the model [46]. When the number of hidden layer nodes is 9, the trained model has the minimum mean squared error (MSE) and the maximum R. Therefore, 9 neurons were selected for the optimal performance of neural network model. In Figure 9, the plotting of MSE versus the epochs number for optimal ANN models was presented.

Figure 10 illustrates a comparison between experimental and predicted results for the fitting in the different stages by the optimum number of hidden layer nodes. The correlation coefficient (R) of the line representing the fitting goodness between the model and the experimental data was 0.9332. These linear fitting results showed that the neural network model could simulate and reproduce the extraction process with high accuracy.

The trained neural network was used to simulate the data of orthogonal experiment, which are shown in Table 4. In Table 4, the predicted value of BP neural network model could well match the experimental value. The results showed that the neural network model could simulate and reproduce the extraction process with high accuracy, thereby predicting the amount of triterpenoid acid extracted from *G. lucidum*. Therefore, we could get a conclusion that BP neural network model could simulate and reproduce the application of ionic liquid in the extraction of triterpenoid acid from *G. Lucidum* with high accuracy.

The weights are coefficients between the artificial neurons, which are analogous to synapse strengths between the axons and dendrites of real biological neurons [30]. In Table 5, the internodes weights produced by the ANN model were used in this work. Among the input variables studied, the rotational speed appeared to be the most influential parameter with relative importance of 21.90%, followed by the ultrasonic extraction power, the ultrasonic extraction time, the solid–liquid ratio, and the ILs concentration with relative importance of 21.80%, 20.19%, 19.21%, and 16.90%, respectively.

### 2.5. Method Validation

The standards were injected 6 times continuously to measure the precision, and the result showed relative standard deviation (RSD) values of 1.68% and 1.38%, respectively, indicating that the instrument has good precision and can accurately reflect the content of the substance.

Moreover, the recovery tests were measured by the standard-addition method at 80% concentration levels. The recoveries were 98.21% and 88.61%, and the RSD values were 2.09% and 1.67%, respectively, indicating that the method is reliable.

The stability was investigated by the determination of two extracts of the samples under the optimized ILUAE procedure, and the RSD values were 0.25% and 0.42%, respectively. 

Six samples were accurately weighed and were prepared according to the above optimal conditions. The RSD values were 4.66% and 3.49%, respectively, which showed that the test method had good repeatability.

## 3. Materials and Methods

### 3.1. Chemicals and Materials

Ganoderic acid A and D (Figure 11) with purity greater than 98% was purchased from Chengdu Pufei De Biotech Co., Ltd. (Chengdu, China).

The 1-butyl-3-methylimidazole bromide ([BMIM]Br; ≥98), 1-butyl-3-methylimidazolium tetrafuoroborate ([BMIM]BF4; ≥98) and 1-butyl-3-methylimidazolium hexafuorophosphate ([BMIM]PF6; ≥98) were obtained from a limited partnership Merck (Darmstadt, German). The 1-Hexyl-3-methlimidazolium bromide ([HMIM] Br; ≥98) and 1-Methyl-3-n-octylimidazolium bromide ([OMIM] Br; ≥98) were obtained from Tokyo Chemical Industry Co., Ltd. (Tokyo, Japan).

Acetonitrile (chromatographic grade ≥99.9) was obtained from the Xilong Scientific Factory (Guangdong, China). Acetic acid was purchased from Tianjin Fu Chen Chemical Reagent Factory (Tianjin, China). Pure water was purchased from Hangzhou WahahaBaili Food Co., Ltd., (Zhejiang, China).

A LC-20AT high-performance liquid chromatography system (Shimadzu, Kyoto, Japan) was equipped with a quaternary gradient low-pressure pump, the CTO-20A column oven, an SPD-M20AUV-detector, a SIL-20 auto sampler and a degasser. TGL-16 type high-speed centrifuge was obtained from the Jiangsu Jintan Zhongda instrument factory (Jiangsu, China). KQ-500DE ultrasonic cleaner was acquired from Kunshan Ultrasonic Instrument Co., Ltd. (Jiangsu, China). The AB135-S 1/10 million electronic balance was purchased from Mettler Toledo Instruments Co., Ltd. (Shanghai, China).

### 3.2. Plant Materials and Sample Preparation

The dry fruiting bodies of *G. lucidum* were purchased in July 2018 from Shandong Province, China, and identified by Professor Changqin Li (Henan University, Kaifeng, Henan Province, China). The medicinal materials were preserved in the National R&D Center for Edible Fungus Processing Technology, Henan University.

### 3.3. Preparation of the Standard Solution

Two standard solutions of ganoderic acid A and D were prepared in methanol at a concentration of 0.62 and 0.218 mg/mL, respectively. Calibration curves were constructed by plotting the peak area versus the extraction yield of the target analytes and were obtained by the analysis of five different concentration levels of the standard solutions in triplicate.

### 3.4. Preparation of Test Sample Solution

The powders of *G. lucidum* (20 meshes) and methanol solutions of the different ILs were placed in a volumetric flask. With an ultrasonic power of 400 w, after ultrasonic extraction for 20 min, the mixture was centrifuged at 4000 r/min for 10 min. Before the HPLC analysis, the supernatant was obtained by filtration through a 0.22 μm organic microporous membrane. Each sample was performed in triplicate. The type of ILs, the concentration of selected IL, the ultrasonic power, the ultrasonic time, the rotational speed, and the solid–liquid ratio were systematically investigated in this experiment.

### 3.5. Chromatographic Conditions

Chromatographic conditions were set as follows: separation column, Agilent Eclipse XDB-C18 (4.6 × 250 mm, 5 μm); the mobile phase, acetonitrile (A) −0.2% acetic acid water (C); gradient elution (0–40 min, 30–51%A, 70–49%C); the column temperature, 30 °C; the flow rate, 0.6 mL/min; the UV detection wavelength, 254 nm; and sample volume, 20 μL. The HPLC chromatograms of the standard solution and the sample extract are shown in Figure 12.

### 3.6. Optimization Extraction Process in *Ganoderma lucidum*

The experimental design method is an important way to solve scientific and engineering problems [47]. The orthogonal experiments of 5 factors and 3 levels were designed by SPSS 19.0 to screen out the optimal extraction conditions of ganoderic acid A and D in *G. lucidum*. The range of each factor level was determined based on the results of the preliminary test, as shown in Table 1. The yields (%) of ganoderic acid A and D were taken as the dependent variables. The extraction yield of target compounds was determined with the following formula:
Yields (mg/g) = mean mass of target compounds in herb samples (mg)/mean mass of the herb samples (g)

### 3.7. Artificial Neural Networks (ANNs)

The ANN modeling is based on the working of the natural neural networks of the human brain [48]. ANN is being used as a powerful tool in predicting the behavior of a particular system, to evaluate the existing one and design a new process [49]. The extraction yield was predicted by the MATLAB mathematical software of neural network. In the present study, a three-layer ANN was used. A log-sigmoid transfer function “logsig” was used at the hidden layer, and a linear transfer function “purelin” was used at the output layer (Figure 13). The Levenberg–Marquardt back-propagation algorithm “trainlm” was used for the network because of the fast convergence speed. The experimental data were randomly divided into three categories (70% for training, 15% for testing, and 15% for verification) and then used to develop a neural network model.

There was no limit to the input data of BP neural network. However, when the units and orders of magnitude of input data were different, the training accuracy of the network was affected. All input and output data were normalized between 0 and 1 to avoid numerical overflows. In this study, the normalization of sample data was performed by the function mapminmax in MATLAB software. Optimizing the number of neurons in the hidden layer is probably the most important step in developing an ANN model [50]. Therefore, the number of neurons tested ranged from 3 to 12, using a series of topologies. The minimum mean square error (MSE) and the maximum correlation coefficient (R) were considered when selecting the best performance of the network

The input variables were the concentration of selected IL, the ultrasonic power, the ultrasonic time, the rotational speed, and the solid–liquid ratio. The extraction yield was used as an output variable.

## 4. Conclusions

In this study, an effective method was established to extract ganoderic acid A and D from *G. lucidum*. After consulting the literature [51], it was found that the effect of ILs on extraction of flavonoids, phenols, saponins, and terpenoids was better than that of traditional solvents. Compared with traditional methods, the present approach obtained higher extraction yield of ganoderic acid A and D. The optimum conditions for ILUAE were determined.

The Levenberg–Marquardt back-propagation algorithm was used to optimize the three-layered neural network to predict the extraction yield of ganoderic acid A and D from *G. lucidum*. The configuration of 9 neurons in the hidden layer led to the MSE (0.09622) and high correlation coefficient (R = 0.93332). For the considered experimental conditions, neural network modeling could effectively simulate the experimental data and reproduce the process behavior.

## Figures and Tables

**Figure 1 molecules-25-01309-f001:**
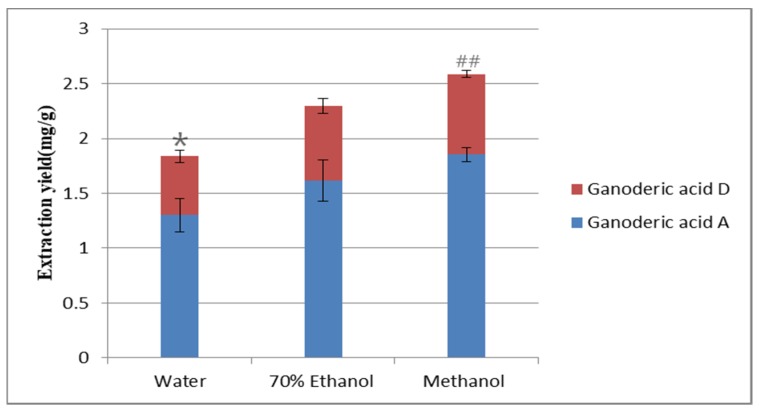
Influence of different extraction solvents. Compared with methanol group: * *p* < 0.05. Compared with [HMIM] Br/methanol group: ^##^
*p* < 0.01.

**Figure 2 molecules-25-01309-f002:**
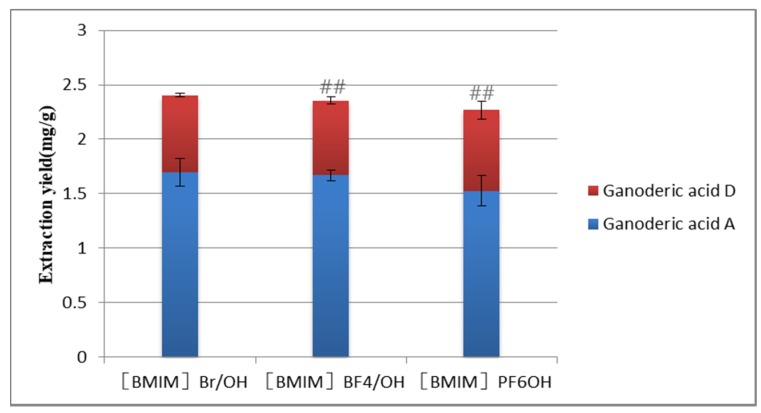
Influence of different types of anions on the extraction yield. Compared with [HMIM] Br/methanol group: ^##^
*p* < 0.01.

**Figure 3 molecules-25-01309-f003:**
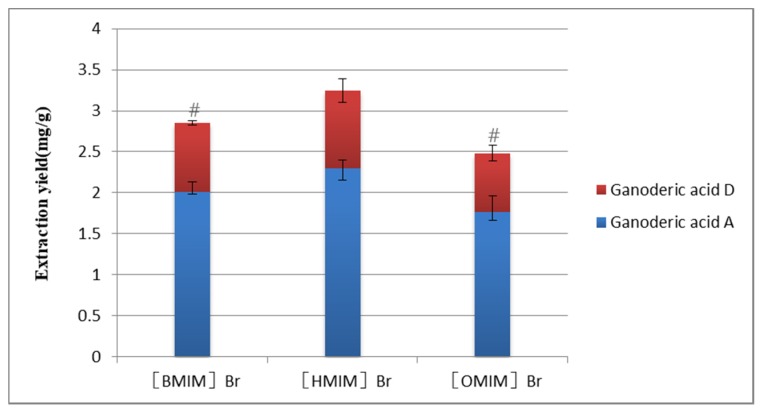
Influence of different types of cations on the extraction yield. Compared with [HMIM] Br/methanol group: ^#^
*p* < 0.05.

**Figure 4 molecules-25-01309-f004:**
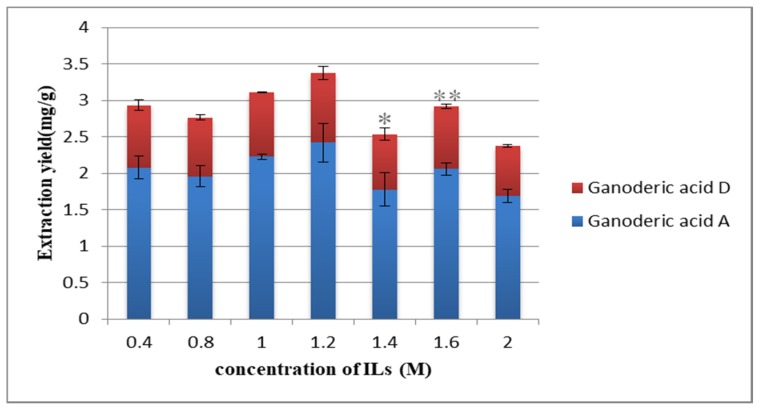
Effect of concentration of ionic liquids (ILs) on the extraction yield. Compared with 1.2M group: ** *p* < 0.01; * *p* < 0.05.

**Figure 5 molecules-25-01309-f005:**
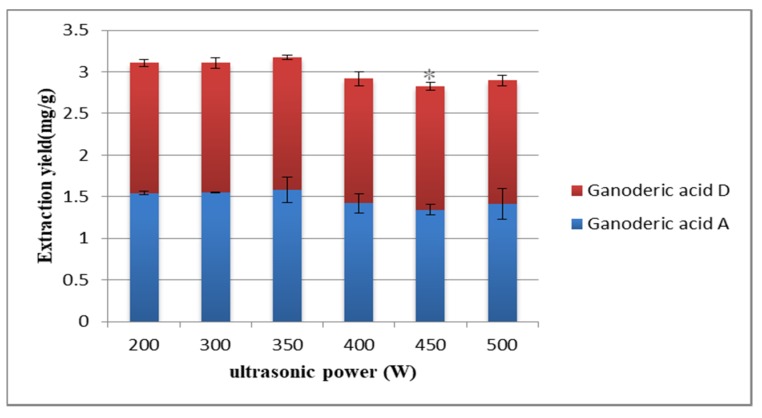
Effect of ultrasonic power on the extraction yield. Compared with 350 W group: * *p* < 0.05.

**Figure 6 molecules-25-01309-f006:**
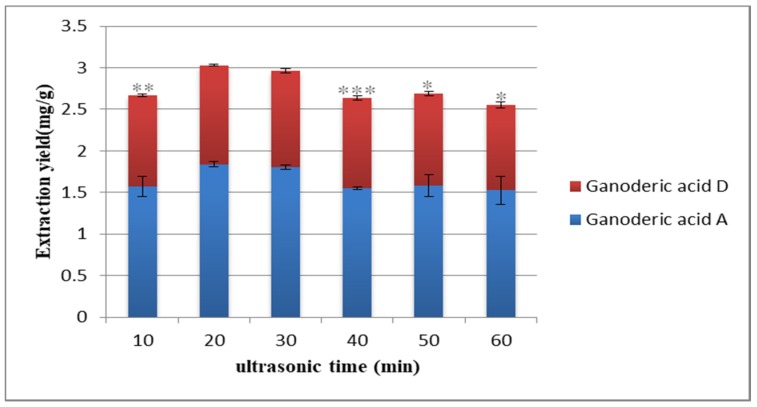
Effect of ultrasonic time on the extraction yield. Compared with 20 min group: *** *p* < 0.001; ** *p* < 0.01; * *p* < 0.05.

**Figure 7 molecules-25-01309-f007:**
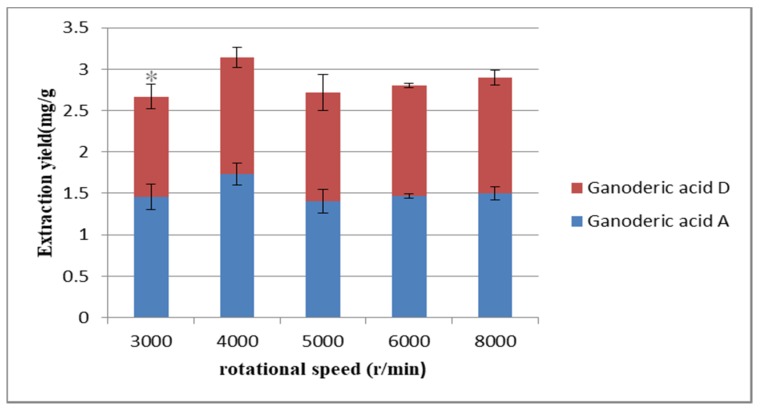
Effect of rotational speed on the extraction yield. Compared with 4000 r/min group: * *p* < 0.05.

**Figure 8 molecules-25-01309-f008:**
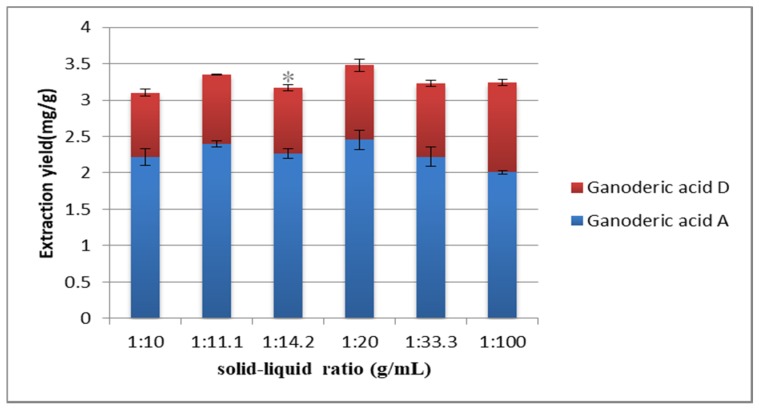
Effect of solid–liquid ratio on the extraction yield. Compared with 1:20 g/mL group: * *p* < 0.05.

**Figure 9 molecules-25-01309-f009:**
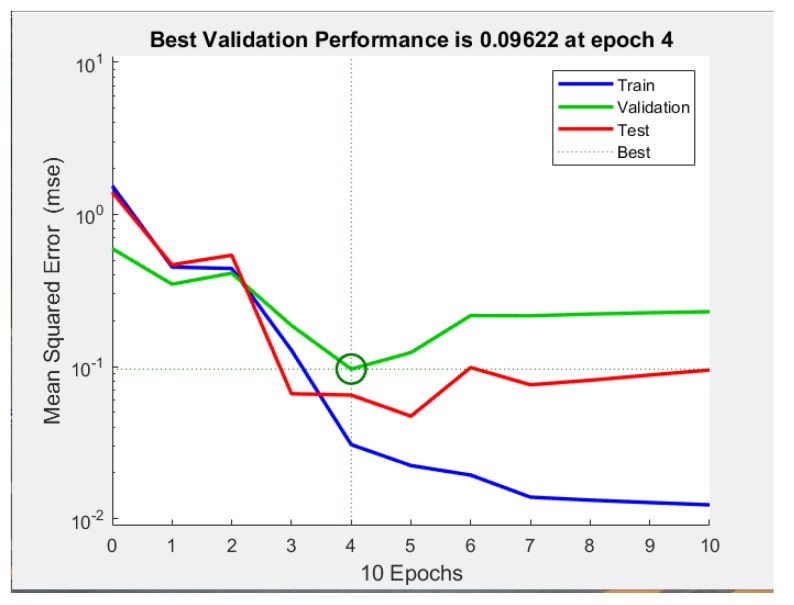
Mean squared error (MSE) versus the number of epochs.

**Figure 10 molecules-25-01309-f010:**
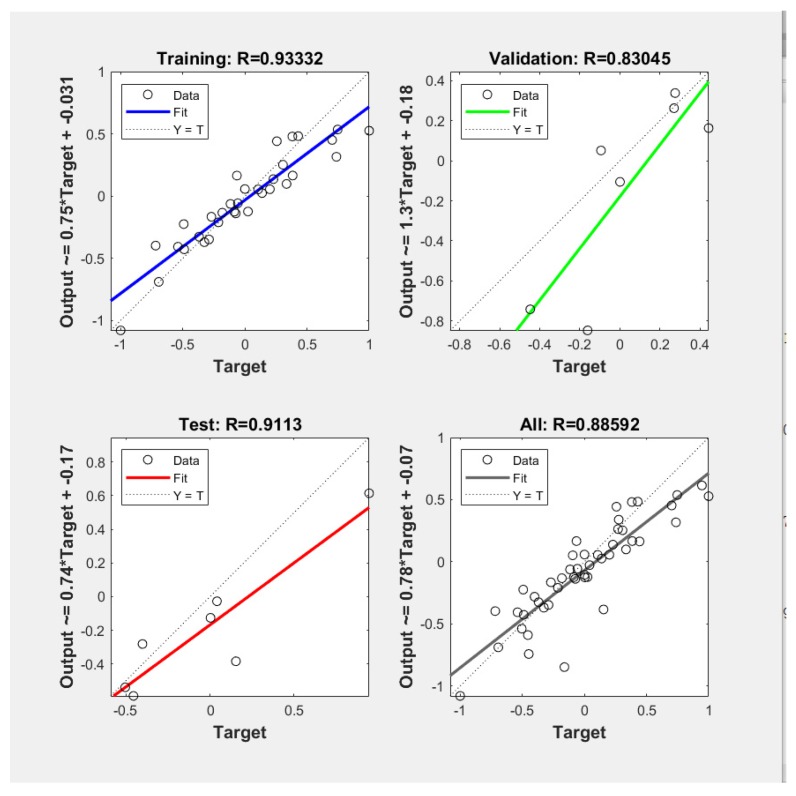
Experimental values versus artificial neural network (ANN) predicted values.

**Figure 11 molecules-25-01309-f011:**
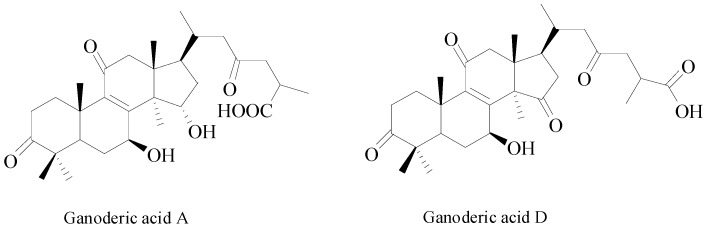
Chemical structures of Ganoderic acid A and D.

**Figure 12 molecules-25-01309-f012:**
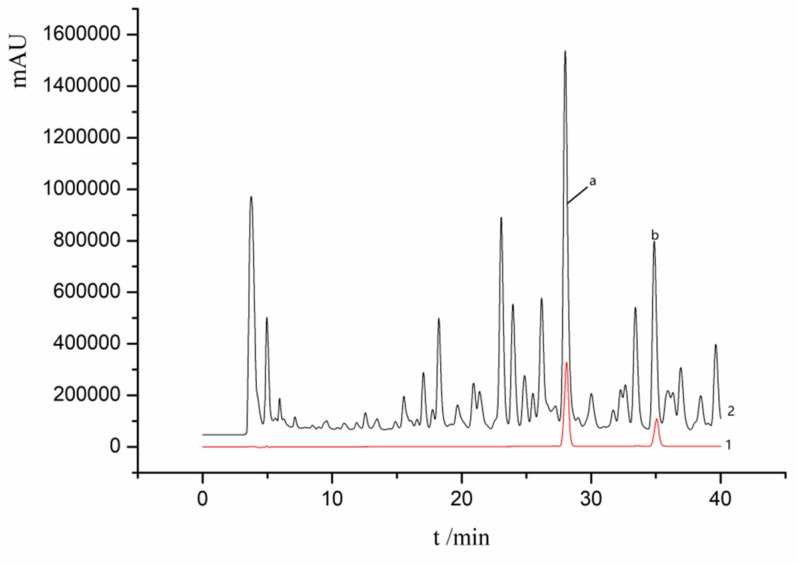
HPLC chromatograms of the standard solution (1) and the test sample solution (2): (a) Ganoderic acid A, (b) Ganoderic acid D.

**Figure 13 molecules-25-01309-f013:**
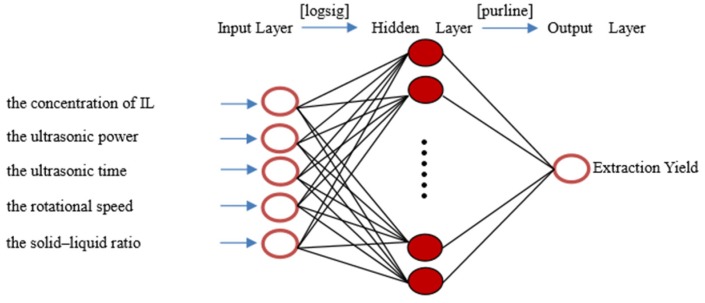
Structure of the developed ANN with five inputs and one output.

**Table 1 molecules-25-01309-t001:** Orthogonal test factors and level tables.

Factor Level	A IL Concentration (mol/L)	B Ultrasonic Time (W)	C Ultrasonic Time (min)	D Rotational Speed (r/min)	E Solid—Liquid Ratio (g/mL)
1	1	300	10	3000	1:14.2
2	1.2	350	20	4000	1:20
3	1.4	400	30	5000	1:33.3

**Table 2 molecules-25-01309-t002:** Results of extreme analysis.

NO.	A	B	C	D	E	F	Extraction Yield mg/g
1	2	3	1	3	3	3	3.013
2	1	3	1	2	2	1	3.040
3	2	2	3	2	3	1	2.951
4	2	1	3	3	2	1	2.953
5	3	1	1	3	3	2	2.908
6	1	2	2	3	2	3	2.795
7	2	2	1	1	1	3	2.692
8	3	2	1	2	2	2	2.719
9	1	1	1	1	1	1	2.660
10	3	2	2	3	1	1	3.083
11	2	1	2	2	1	2	3.067
12	2	3	2	1	2	2	2.784
13	1	3	3	3	1	2	2.975
14	3	1	3	1	2	3	2.738
15	1	2	3	1	3	2	2.846
16	3	3	3	2	1	3	3.198
17	3	3	2	1	3	1	2.951
18	1	1	2	2	3	3	3.207
K1	17.523	17.533	17.032	16.673	17.675		
K2	17.459	17.087	17.887	18.182	17.029		
K3	17.598	17.961	17.661	17.726	17.876		
k1	5.841	5.844	5.677	5.558	5.892		
k2	5.82	5.696	5.962	6.061	5.676		
k3	5.866	5.987	5.887	5.909	5.959		
R	0.046	0.291	0.285	0.503	0.067		

**Table 3 molecules-25-01309-t003:** Variance analysis of factor.

Source	Type III Sum of Squares	df	Mean Square	F	Sig.
A	0.002	2	0.001	0.075	0.929
B	0.064	2	0.032	3.045	0.112
C	0.065	2	0.033	3.122	0.107
D	0.200	2	0.100	9.557	0.010
E	0.065	2	0.033	3.115	0.108
Error	0.073	7	0.01		
Total	154.062	18			

**Table 4 molecules-25-01309-t004:** Prediction results of BP neural network.

NO	Observed Value	Predicted Yield	Relative Error %
1	3.013	2.983	0.98
2	3.040	2.729	10.22
3	2.951	2.984	1.12
4	2.953	2.878	2.52
5	2.908	2.871	1.28
6	2.795	2.856	2.17
7	2.692	2.523	6.29
8	2.719	2.788	2.52
9	2.660	2.640	0.76
10	3.083	3.030	1.72
11	3.067	2.983	2.73
12	2.784	2.750	1.23
13	2.975	2.935	1.33
14	2.738	2.763	0.9
15	2.846	2.874	0.98
16	3.198	3.230	0.99
17	2.951	2.890	2.07
18	3.207	3.045	5.05

**Table 5 molecules-25-01309-t005:** The relative weight of five factors and extraction yield.

Factor	Weight (%)
A	16.90
B	21.80
C	20.19
D	21.90
E	19.21

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
