# Peer review of "Ionic Liquid-Based Ultrasonic-Assisted Extraction Coupled with HPLC and Artificial Neural Network Analysis for Ganoderma lucidum"

_molecules, 2020, doi:10.3390/molecules25061309_

Round 1
Reviewer 1 Report
In the introduction part data regarding this neuronal networks should be included.
Line 33- As –unbold and delete the sign “;” and replce with “,”
Line 40: Delete Wu et all, there is already the citation at the end
Line 266-the authors collected directly the dry fruit? Please specify
Line 270-271 is contradictory with line 64-71
The mainly goal of this article is the comparing the results with other literature data. For sure, regarding Figure 1 in which the most used solvents (water, ethanol and methanol) are used in the literature. And the ionic liquids used as a possible alternative does/t not have been used in the literature until now? Maybe on other matrices. The same question for using the selection of ultrasonic power, time or speed. In this part the authors doesn’t compare with the literature data.
If I understood well this neuronal network is used in order to predict the best process conditions for extracting ganoderic acids A and D. But I could not find and understood the prediction was supported by the results obtained? The authors must conclude this clearly in the text.
Author Response
- In the introduction part data regarding this neuronal networks should be included.
Answer: Thanks for your suggestion. We have added neural network data in the introduction part. Please check it.
- Line 33: As -unbold and delete the sign“;”and replce with“,”.
Answer: Thank you for your suggestion. We have corrected them in the manuscript. Please check it.
- Line 40: Delete Wu et all, there is already the citation at the end.
Answer: Thank you for your suggestion. We deleted “Wu et all”. Please check it.
- Line 266: the authors collected directly the dry fruit? Please specify.
Answer: Thank you for your suggestion. The dry fruiting body of G. lucidum were purchased from Shandong Province, China. We have added this. Please check it.
- Line 270-271 is contradictory with line 64-71.
Answer: Thank you for your suggestion. We have added the description of the preparation of the standard solution in Section 3.3. Please check it.
- The mainly goal of this article is the comparing the results with other literature data. For sure, regarding Figure 1 in which the most used solvents (water, ethanol and methanol) are used in the literature. And the ionic liquids used as a possible alternative does/t not have been used in the literature until now? Maybe on other matrices. The same question for using the selection of ultrasonic power, time or speed. In this part the authors doesn't compare with the literature data.
Answer: Thank you for your suggestion. We have enriched the “Results and discussion” and compared with literatures donated in red. Please check it.
- If I understood well this neuronal network is used in order to predict the best process conditions for extracting ganoderic acids A and D. But I could not find and understood the prediction was supported by the results obtained? The authors must conclude this clearly in the text.
Answer: Thank you for your suggestion. The ANN model development was established based the data of single factor experiments and orthogonal experiment. The correlation coefficient (R) of the line representing the fitting goodness between the ANN model and the experimental data was 0.9332. The trained neural network was used to simulate the data of orthogonal experiment, which shown in Table 4. The predicted value of BP neural network model could well match the experimental value. Therefore, we could get a conclusion that the neural network model could simulate and reproduce the extraction process with high accuracy, and thereby predicting the amount of triterpenoid acid extracted from G. lucidum. All these descriptions were added in Section 2.5.

Reviewer 2 Report
The science of the manuscript is fine,
Please indicate results of statistical analysis in each figure.
The manuscript needs a great deal of language polishing.The following shows only some examples. e.g.line 207 illustrates. line 215 are shown,line 252 acetonitrile was. line 254 pure water,line 261 materials were,line 266 bodies,line 285 are shown,line 293 yields, line 301 was,NOT was be
A variety of formats is found in the reference list.Please be consistent.Please refer to references #10,27,28,32,36,34,37.
Author Response
Response to reviewer 2:
- The science of the manuscript is fine. Please indicate results of statistical analysis in each figure.
Answer: Thank you for your suggestion. We have added the statistical analysis in each figure. Please check related pictures.
- The manuscript needs a great deal of language polishing. The following shows only some examples. e.g. line 207 illustrates; line 215 are shown; line 252 acetonitrile was; line 254 pure water; line 261 materials were; line 266 bodies; line 285 are shown; line 293 yields; line 301 was not was be.
Answer: Thank you for your suggestion. We have revised these questions. Please check it.
- A variety of formats is found in the reference list. Please be consistent. Please refer to references 10, 27, 28, 32, 36, 34, 37.
Answer: Thank you for your suggestion. We have checked all the formats of the reference. Please check it.

Round 2
Reviewer 2 Report
The revised manuscript is acceptable for publication.